# Thermotropic Liquid Crystalline Polymers with Various Alkoxy Side Groups: Thermal Properties and Molecular Dynamics

**DOI:** 10.3390/polym11060992

**Published:** 2019-06-04

**Authors:** Gi Tae Park, Jin-Hae Chang, Ae Ran Lim

**Affiliations:** 1Department of Polymer Science and Engineering, Kumoh National Institute of Technology, Gumi 39177, Korea; pkgt0129@gmail.com; 2Analytical Laboratory of Advanced Ferroelectric Crystals, Jeonju University, Jeonju 55069, Korea; 3Department of Science Education, Jeonju University, Jeonju 55069, Korea

**Keywords:** thermotropic liquid crystalline polymers, thermal property, ^13^C cross-polarization/magic angle spinning, ^13^C spin-lattice relaxation time, molecular dynamics

## Abstract

Two series of thermotropic liquid-crystalline polymers (TLCPs) were synthesized by reacting various dialkoxy terephthalate units with hydroquinone (HQ) and 2,6-naphthalene diol (Naph). The dialkoxy terephthalate moieties used in this study include 2,5-diethoxyterephthalate, 2,5-dibutoxyterephthalate, and 2,5-dihexyloxy-terephthalate. All the TLCPs synthesized in this study formed nematic phases. The molecular motions according to the length of the dialkoxy side groups in the TLCPs were evaluated by ^13^C cross-polarization/magic angle spinning nuclear magnetic resonance spectroscopy. The thermal properties and molecular dynamics of the TLCPs are found to be affected by the length of the dialkoxy side group and the aromatic diol unit in the main chain. Further, the thermal behaviors, liquid crystalline mesophases, and degree of crystallinity of the two series of TLCPs, i.e., HQ- and Naph-TLCPs, are compared.

## 1. Introduction

Liquid crystals (LCs) are materials that exhibit partially ordered states, with their molecular order lying between those of three-dimensionally ordered crystals and disordered or isotropic liquids. As a consequence of the molecular order, LC phases are anisotropic, i.e., they have directional properties. Formation of LC states is a consequence of either structures with rigid-rod type units with a high axial ratio or of disc-like structures. These structural units are often called mesogenic units. The LC states of polymers were discovered with the discovery of aramids such as poly(*p*-phenylene terephthalamide) (Kevlar) and poly(*p*-benzamide) by DuPont de Nemours Co. in 1970s [1,2]. These aromatic polyamides form LC states when dissolved in a solvent (lyotropic) such as sulfuric acid. In addition, commercialization of aromatic polyesters (e.g., Xydar^®^ and Vectra^®^) that form LC states in melts (thermotropic) in the 1980s has sparkled continued and unabated growth of the field of LC polymers (LCPs) [3,4].

Thermotropic liquid crystalline polymers (TLCPs) represent a class of high-performance polymers that have excellent thermo-mechanical properties combined with a low density, resulting in excellent specific properties [5,6]. This unique feature of TLCPs arises from the rigidity of the polymer backbone and/or the pendant groups on the main polymer chain in symmetric rigid sequences that are classified as mesogens [7,8,9]. They are typically applied in electronic devices, protective coatings, and sports equipment, as automotive part composites, etc. [10,11,12]. 

Many TLCPs contain a completely aromatic backbone, which maximizes their thermo-mechanical properties [13,14]. The mesogenic structures are mainly synthesized from terephthalic acid (TPA), 2,6-naphthalene-dicarboxylic acid, *p*-hydroxybenzoic acid (*p*-HBA), hydroquinone (HQ), 2,6-naphthalene diol (Naph), and *p*,*p*′-biphenol in their unsubstituted or substituted forms [11,13,14,15,16]. However, rigid rod-type TLCPs often have very high melting points, which make their processing difficult. Such TLCPs also have low solubility in common solvents [17].

Wholly aromatic LCPs, especially when they are homopolymers, are highly crystalline, very-high melting, and very often interactive materials. Two representative examples are poly(*p*-hydroxybenzoate) and poly(*p*-phenylene terephthalate). These polymers have a very high melting temperature, and thus, they cannot be readily processed by melt spinning or injection molding. In contrast, copolyesters such as those (Xydar produced from Amoco, Chicago, Illinois, USA) prepared from *p*-HBA, TPA, and *p*,*p*′-biphenol are injection moldable at temperatures in the vicinity of 400 °C, which is not compatible with common melt spinning equipment. Thermal degradation of the copolyester at 400 °C, however, makes stable fiber production difficult. The other classes of wholly aromatic LCPs contain bent structural moieties such as resorcinol and bisphenol-A in addition to para-linked aromatic structures. Inclusion of asymmetrically substituted units such as chloro- or methylphenylene units can further depress the melting temperature. Further, the class of aliphatic-aromatic LCPs consists of mesogenic groups sequenced in a regular fashion with flexible, aliphatic segments of different lengths [18,19,20].

However, TLCPs should be structurally modified to lower their melting temperatures for overcoming the processing issues and prevent their thermal degradation before melting. The most common method for structural modification is to combine different mesogenic monomers such as bulky side substituents, flexible alkyl side groups, or kink (non-linear)-structured monomers [21,22,23]. Lenz et al. [3] demonstrated a shift in the transition temperatures to lower values by attaching substituents with different structures and varying chain lengths to the main polymer chain, which allowed a systematic study of the processing of TLCP systems. Griffin and Kricheldorf [24,25] used kinked monomers such as *m*-hydroxybenzoic acid (*m*-HBA) or various diols; these researchers lowered the melting temperatures of TLCPs by including flexible alkyl groups into the polymer main chain. In addition, various studies have been carried out using TLCPs having alkyl side chains. Recently, microphase separation was studied using an LC copolymer containing alkyl side groups [26,27,28].

Further, ^13^C nuclear magnetic resonance (NMR) has proven to be a powerful technique for studying the local dynamic motions of molecules [29]. The ^13^C spin-lattice relaxation time (*T_1ρ_*) in the rotating frame is an important experimental variable for probing the dynamic processes of a molecule [30]. By studying the relaxation of the nuclei in different environments of the chain, it is possible to obtain a detailed picture of the motion of different parts of the chain in a polymer. The measured relaxation data can be used to obtain information regarding the dynamic processes occurring in different parts of the chain.

In this study, the thermal and physical properties of two new TLCP series were evaluated. TLCPs with dialkoxy side groups were synthesized from dialkoxy-TPA and two diols, HQ and Naph. The thermal properties of the two series of TLCPs were then studied comparatively. The characteristics of each TLCP were compared according to the variation in the side group length, with the number of carbons in the dialkoxy group varied as 2, 4, and 6.

In addition, the structures of the TLCPs with various alkoxy side groups were confirmed by ^13^C NMR spectroscopy. The spin-lattice relaxation time, *T_1ρ_*s, in the rotating frame for each carbon of the dialkoxy side group and main chain in the HQ- and Naph-TLCPs was measured to understand the molecular dynamics. The effect of the length of the dialkoxy side groups on the carbon mobility was evaluated. In addition, the molecular motions according to the length of the dialkoxy groups in the TPA moiety were compared using the relaxation times.

## 2. Experimental

### 2.1. Synthesis of 2,5-Dialkoxylterephthaloyl Chloride

The chemical structures of the precursors are shown in Figure 1. 2,5-Dialkoxyterephthaloyl chloride (product C in Figure 1) was synthesized by the reaction of dialkyl 2,5-dihydroxy-terephthalate with the respective normal alkyl bromide in multiple steps [22].

### 2.2. Synthesis of TLCPs

All TLCPs were prepared in solutions. The synthetic procedures for all TLCPs are similar. Herein, the preparation of 2,5-diethoxyterephthaloyl chloride/HQ TLCP is provided as a representative example. In a typical synthesis, a solution of 0.2 mol (58.226 g) of 2,5-diethoxyterephthaloyl chloride (C in Figure 1) and an equivalent amount of HQ in 600 mL of tetrachloroethane (TCE) was taken in a three-neck flask fitted with a dropping funnel and magnetic stirrer. Then, an excess of pyridine was added slowly at room temperature under N_2_ and the mixture was stirred at 120 °C for 14 h. Thereafter, the solution was diluted and poured into excess methanol. The precipitated product was filtered and washed several times with methanol, and then dried in vacuum. The reaction route is shown in Figure 1. The inherent viscosity of the resulting TLCPs in phenol/*p*-chlorophenol/TCE = 25/40/35 (w/w/w) is 0.83–0.99 dL/g, as measured at a concentration of 0.1 g/dL (see Table 1). The products of diethoxy-TPA, dibutoxy-TPA, and dihexyloxy-TPA with HQ are denoted as HQ-2, HQ-4, and HQ-6, respectively, and the products of diethoxy-TPA, dibutoxy-TPA, and dihexyloxy-TPA with Naph are denoted as Naph-2, Naph-4, and Naph-6, respectively. All chemical structures are shown in Figure 2.

### 2.3. Characterization

Differential scanning calorimetry (DSC) and thermogravimetric analysis (TGA) of the TLCPs were carried out under a N_2_ atmosphere using DuPont 910 equipment (DuPont, New Castle, DE, USA). The samples were heated and cooled at the rate of 20 °C/min. 

A polarizing microscope (Leitz, Ortholux) (Lahn-Dill-Kreis, Wetzlar, Germany) equipped with a Mettler FP-5 hot stage was used to examine the liquid crystalline behavior of the TLCPs. The photographs of the liquid crystals were obtained as follows: A small amount of the sample was placed on a slide glass and the nematic mesophase was observed by slowly increasing the temperature after covering it with a cover glass. The liquid crystalline phase was obtained at *T_m_* or more and *T_i_* or less, and the magnification was 100×. LC photographs were obtained after heat treatment at a constant temperature for 30 min to better observe the nematic structure for all samples. However, in the case of HQ-4, heat treatment was performed for 2 h.

Wide-angle X-ray diffraction (WAXD) was performed at room temperature on a Rigaku (D/Max-IIIB) X-ray diffractometer (Rigaku, Tokyo, Japan) using Ni-filtered Cu-*Kα* radiation. The patterns were recorded over the range of 2*θ* = 2°–35° at a scanning rate of 2°/min. The ChemDraw program was used for generating the 3D chemical structures.

### 2.4. NMR Spectroscopy

The ^13^C chemical shifts and *T_1ρ_* for the HQ- and Naph-TLCPs were obtained by ^13^C cross-polarization/magic angle spinning (CP/MAS) NMR at Larmor frequencies of ω_0_/2π = 100.61 MHz using Bruker 400 MHz NMR spectrometers at the Korea Basic Science Institute, Western Seoul Center. The chemical shifts were referenced to tetramethylsilane (TMS). Powdered samples were placed in a 4-mm CP/MAS probe and the MAS rate was set to 10 kHz for all ^13^C measurements to minimize the spinning sideband overlap. The ^13^C *T_1ρ_* values were measured by varying the duration of the ^13^C spin-locking pulse applied after the CP preparation period. The width of the π/2 pulse used for measuring *T_1ρ_* of ^13^C was 3.3 μs. The decay of the ^13^C magnetization in the spin-locking field was followed for spin-locking times of up to 200 ms.

## 3. Results and Discussion

### 3.1. Thermal Properties of the TLCPs

The DSC thermograms of the TLCPs containing HQ and Naph moiety are shown in Figure 3. The thermal properties of HQ-TLCPs were compared with those of the Naph-TLCPs. All the TLCPs exhibited a clear glass transition temperature (*T_g_*), melt transition temperature (*T_m_*), and isotropic transition temperature (*T_i_*), regardless of the structure of the different dialkoxy mesogenic units.

The key differences in the structural units of the TLCPs under investigation are based on different lengths of the dialkoxy side groups. For the HQ-TLCPs, the *T_g_* decreased with increasing chain length of the dialkoxy side groups, and HQ-2 showed the highest *T_g_* (71 °C), which is 60 °C higher than that of HQ-6 (11 °C), as shown in Table 1. This could be because of the higher rigidity of the diethoxy side groups than that of the dihexyloxy substituents in the terephthaloyl moiety. The *T_g_* of a polymer is closely related to the flexibility of its chain; a high *T_g_* is generally attributed to relatively high energy barriers to bond rotations. The increase in *T_g_* is likely due to two factors: (1) the significant effect of the rigid rod-like monomer structure on the free volume of the polymer, and (2) the confinement of the inter-reacted polymer chains within the polymers, which prevent the segmental motion of the chains [31,32]. 

The low *T_g_* (11 °C) of the HQ-6 TLCP is possibly because the long dialkoxy substituents disturb its linearity and thus increase the flexibility of its backbone. It is generally accepted that the introduction of a long alkoxy side group into a polymer structure deteriorates its thermal properties [31].

The *T_m_* and *T_i_* of the HQ-TLCPs containing diethoxy-side groups were also found to be higher than those of the TLCPs containing dibutoxy and dihexyloxy side groups. For example, *T_m_* and *T_i_* decreased gradually from 279 to 164 °C and 325 to 200 °C, respectively, with an increase in the chain length of the alkoxy groups from ethoxy to hexyloxy. This trend is attributed to the length of the dialkoxy side groups in the TLCP structures. The TLCP containing a diethoxy group has a more linear structure, whereas the TLCPs containing dibutoxy- and dihexyloxy-side groups are non-linear. The interlocking effect of the diethoxy units favors linearity of the molecular shape in the mesophase formation, whereas similar interlocking effects are not anticipated from the long alkoxy-substituent structures. This effect could explain why the thermal transition temperatures of the structures with diethoxy group are higher than those of the structures with dibutoxy and dihexyloxy groups [31,32,33].

Similar thermal behaviors were observed for the Naph-TLCPs. In particular, Naph-2 is composed of two continuous benzene rings and the shortest ethoxy substituents, and has better thermal stability than other structures. For example, the *T_g_*, *T_m_*, and *T_i_* of Naph-TLCPs decreased from 76 to 38 °C, 282 to 164 °C, and 329 to 227 °C, respectively, with an increase in the side chain length of the side groups from ethoxy to hexyloxy, as seen in Table 1. These results suggest that introducing various alkoxy substituents into rigid rod-type main-chain TLCPs could diminish their thermal properties and even the problems such as their decomposition at high temperatures during their processing can thus be solved. 

The TGA thermograms of the two series of TLCPs are shown in Figure 2; the curve in the temperature range of 300 to 400 °C is enlarged in the inset of Figure 4. The comparative TGA results for the HQ- and Naph-TLCPs with various side groups are also listed in Table 1. The data in Table 1 indicate that the initial thermal degradation temperature (*T*_D_^i^) of the two series decreases with an increase in the length of the alkoxy substituents, as they show the same trend in *T_g_*, *T_m_*, and *T_i_*. Further, *T*_D_^i^ at 2% weight loss varied from 357 to 342 °C as the chain length of the alkoxy group in the side substituents increased from ethyl to hexyl. This decrease in the thermal stability could be attributed to the low thermal stability of the alkoxy group. The weight of the residue at 600 °C (wt_R_^600^) of the HQ-series TLCPs decreased from 27 to 20% as the alkoxy length increased from ethoxy to hexyloxy (see Table 1). This decrease in the char formed is because of the low heat of resistance of the alkoxy moiety. Table 1 also lists the thermal stabilities of the Naph series with various alkoxy lengths. *T*_D_^i^ of the Naph series also decreases from 365 °C to 348 °C as the chain length of the alkoxy group in the side substituents increased from ethyl to hexyl. The wt_R_^600^ for the Naph-TLCPs also decreases linearly from 48% to 26% as the alkoxy substituent chain length increases up to hexyloxy. 

It is evident from Table 1 that all the thermal properties (*T_g_*, *T_m_*, *T_i_*, *T*_D_^i^, and wt_R_^600^) of the Naph-TLCPs are better than those of the HQ-series TLCPs. The high axial ratio (length to diameter) of the Naph-TLCPs resulted in higher transition temperatures than those of the HQ-TLCPs. This could be explained by the presence of more phenyl rings in the main chain of Naph-TLCPs [34,35]. 

To understand the origin of the differences in the thermal properties of the HQ- and Naph-TLCPs, the molecular geometry of HQ-6 and Naph-6 are shown in Figure 5. A comparison of the molecular shape of the two series reveals that the Naph-6 chain is more linear when compared with the chain structure of the HQ-6 conformer. It is also evident that the degree of bending and agglomeration of the HQ-6 polymer is more severe than that of the Naph-6 polymer. The specific chain conformations of all HQ- and Naph-TLCPs appear to lead to the same degree of linearity of the molecular shape (see Figure 5). The possible interlocking of the extruding Naph-series units favors mesophase formation, whereas a similar interlocking effect is less likely for the HQ-series TLCPs.

### 3.2. LC Mesophases

All the TLCPs containing the HQ- and Naph moiety formed LC mesophases above the respective *T_m_* or in the fluid melt state, regardless of the length of the dialkoxy side groups [36,37,38]. This implies that when a polyester chain consists on average of one diol unit and one terephthaloyl group, i.e., two consecutive linear monomeric structures in its repeating unit, it is able to form a liquid crystalline mesophase even when a long dialkoxy side group, such as the hexyloxy moiety, is included in the main chain. 

Typical optical micrographs of the optical textures of the LC structures obtained using a polarizing microscope are shown in Figure 6. All the TLCPs with different lengths of dialkoxy groups exhibited a threaded Schlieren texture typical of nematogens. The mesogenic LC textures of the Naph-series TLCPs exhibited a fine mesophase texture perpendicular to the shear direction and showed more threaded and developed textures than those of the HQ-series TLCPs. Based on these results, we conclude that the molecular geometry of the naphthalene unit affects the LC texture (see Figure 5) [38]. 

The stability of the mesophase is affected by the rigidity and aspect ratio of the mesogenic molecule. The aromatic rigid-rod mesogens stabilize the mesophase of the formed LC, owing to the increase in the geometrical anisotropy [39,40]. The ester linkage stabilizes the mesophase of the TLCP through mutual conjugation between the ester units, regardless of the length of the dialkoxy side groups. 

### 3.3. WAXD Analyses 

The WAXD patterns of the TLCPs are shown in Figure 7. Although their crystal structures are different, the X-ray diffraction patterns of these TLCPs have common crystalline features: strong and sharp peaks between 2*θ* = 6 and 2*θ* = 26°, indicating a stable semi-crystalline character. Further, the degree of crystallinity (DC) is calculated as, Crystallinity% = [*Ic*/(*Ic* + *Ia*)] × 100, where *Ic* is the intensity of the crystalline area, and *Ia* is the intensity of the amorphous area [41]. The approximate DCs estimated from the diffractograms are listed in Table 1. 

In Figure 7, the intensity and shape of the peaks are very different depending on the alkoxy chain length in HQ- and Naph-series TLCPs. However, the peak pattern and peak intensity are similar for the HQ- and Naph series at the same length of the dialkoxy side groups. For example, HQ-2 and Naph-2 show strong peaks at almost similar positions because of their short side groups. Likewise, similarly results are obtained for HQ-4 and Naph-4, and for HQ-6 and Naph-6. Overall, HQ-6 and Naph-6 with dihexyloxy side groups show an amorphous peak in the diffractograms compared to those of HQ-2 and Naph-2 with ethoxy side groups. This is because the length of the alkyl side group has a significant effect on the crystallinity.

Compared with that of HQ-4, the sharper peak at 2*θ* = 8.32° (*d* = 10.63 Å) in the diffraction pattern of Naph-4, indicates that it has higher crystallinity (24%) and a different crystal structure. Therefore, the dibutoxy unit in Naph-4 favors linearity in mesophase formation, whereas a similar molecular mesophase effect in the HQ-4 structure (19%) is not anticipated. The TLCP containing a dibutoxy group in the Naph moiety has a more linear structure, whereas the HQ-TLCP containing the dibutoxy side group is non-linear. The DC of the HQ-series TLCPs decreased from 25 to 18% with increasing length of the dialkoxy side groups. However, the DC of the Naph-series TLCPs ranged between 20 and 30% depending on the dialkoxy moieties (see Table 1). Based on the results in Figure 7, the *d*- and 2*θ*-values of each XRD peak are summarized in Table 2. 

In order to investigate the change in the morphological behavior, XRD patterns were obtained at each temperature showing the LC mesogenic phase (see Figure 8). The XRD patterns obtained between *T_m_* and *T_i_* are very different from those obtained at room temperature. That is, in the molten state, all the sharp peaks observed at room temperature disappeared and only a broad peak is observed. As a result, in the molten state, only an amorphous ordered structure formed. 

### 3.4. ^13^C Chemical Shifts and Relaxation Times 

Structural analyses of the dialkoxy side group and main chain in HQ- and Naph-series TLCPs were carried out by solid state ^13^C CP/MAS NMR. The ^13^C chemical shifts of the HQ-series TLCPs were obtained at room temperature. The chemical shifts for CH_3_, CH_2_, CH_2_, CH_2_, CH_2_, and CH_2_ for HQ-6 are observed at 14.23, 22.96, 26.16, 30.09, 31.99, and 68.77 ppm, respectively, as shown in Figure 9. The spinning sidebands are marked with an asterisk. Here, the signals at 151.87, 148.99, 123.56, and 116.78 ppm are assigned to the phenyl ring and HQ moiety in the main chain, and the chemical shift of C=O is 164.98 ppm. The resonance peak for the carbon of the C=O bond has a weaker intensity. The chemical shifts of all carbons are consistent with the structures shown in Figure 9. 

In addition, the ^13^C chemical shifts for the phenyl ring and HQ moiety in the main chain and C=O in HQ-2 and HQ-4 are similar to those of HQ-6, and the chemical shifts for the side groups differed according to the length of the alkoxy groups.

For the Naph-series TLCPs, the peaks of ^13^C in CH_3_, CH_2_, CH_2_, CH_2_, CH_2_, and CH_2_ for Naph-6 are observed at 14.02, 22.69, 26.08, 29.96, 31.71, and 68.98 ppm, respectively, as shown in Figure 10. The ^13^C signal at 164.97 ppm corresponds to C=O, and the signals at 151.96, 148.66, 131.94, 129.03, 122.16, and 117.09 ppm are attributable to the phenyl ring and naphthalene moiety in the main chain. The spinning sidebands are marked with an asterisk. In addition, the ^13^C chemical shifts for phenyl, naphthalene, and C=O in Naph-2 and Naph-4 are similar to those of Naph-6, and the chemical shifts for the alkoxy groups differed based on the length of the alkoxy group.

The *T_1ρ_* values for each carbon of the HQ- and Naph-series TLCPs were measured at room temperature. The traces obtained for carbons in the HQ- and Naph-TLCPs are fitted with a single exponential function, M_z_(*t*)/M(0) = a exp(-*t*/*T_1ρ_*), where M_z_ and M(0) are the loss of magnetization and total nuclear magnetization of ^13^C at thermal equilibrium, respectively [42,43]. From the results, the *T_1ρ_* values for various carbons are listed in Table 3. The relaxation times of carbon in the HQ- and Naph-TLCPs are compared according to the length of the dialkoxy groups. The *T_1ρ_* values of the -CH_3_ and -CH_2_- groups of HQ-2 are 30.7 and 3.1 ms, respectively (Table 3). Further, the *T_1ρ_* values of -CH_3_ in HQ-4 and HQ-6 are longer than the *T_1ρ_* values of other alkoxy groups, -CH_2_-. This suggests that the side groups have additional mobility from internal rotation. The CH_3_ group has a longer relaxation time than other carbons of the side groups, which is consistent with the fact that dipolar relaxation is more efficient based on the number of bonded protons. 

In general, longer *T_1ρ_* values are observed for the carbons of the alkoxy groups attached to the main chain. This results from the greater mobility of the side substituents at the free end. The *T_1ρ_* values for the Naph-TLCPs show a similar trend as those of HQ-TLCPs. The *T_1ρ_* values for the HQ- and Naph-TLCPs represent mobility according to the length of the dialkoxy groups. The dialkoxy groups of the HQ-TLCPs have higher mobility than those of the Naph-TLCPs. In addition, the carbonyl carbons (C=O) have longer *T_1ρ_* than any of the carbon of the dialkoxy side group and main chain. The *T_1ρ_*s of carbons i and j in HQ-TLCP (see Figure 9) and carbon l and m in Naph-TLCP (see Figure 10) are longer than the relaxation times of other carbons in the main chain, as shown in Table 3. The longer *T_1ρ_* values indicate a higher rigidity for main chains in HQ-TLCPs and Naph-TLCPs, which is ascribed to strong interfacial interactions between the alkoxyl groups and main chains.

## 4. Conclusions

In this study, the thermal properties and molecular dynamic behaviors of new TLCPs derived from HQ or Naph and various dialkoxy terephthalate units by solution polymerization were evaluated. All the synthesized TLCPs are thermotropic and nematic, indicating that the linear and rigid aromatic esters are sufficiently long enough to induce mesophase formation. The thermal property and degree of crystallinity decreased with the increasing length of the dialkoxy side groups in the terephthaloyl moiety. The various alkoxy substituents in the rigid rod-type main-chain TLCPs diminished the thermal properties of the TLCPs and thus helped resolve the processing problems associated with their decomposition at high temperatures. These results show that the processing temperature is dependent on the loading level of the monomer containing the side group, with a linear dependency. The thermal properties, liquid crystalline mesophase textures, and degree of crystallinity of the Naph-series TLCPs are better than those of the HQ-series TLCPs, at the same length of the dialkoxy side groups. 

Each carbon was distinguished based on the ^13^C CP/MAS NMR chemical shifts, and the effect of the length of the dialkoxy side groups on the carbon mobility was evaluated. The motions of the dialkoxy groups were enhanced at the opposing ends of the main chain because this group was bound to the aromatic moiety through C=O bonds. The ^13^C *T_1ρ_* was dominated by the fluctuation of the anisotropic chemical shift, and it became longer with larger-amplitude molecular motions. This implies that the amplitude of the motion of the alkoxy group is enhanced at the C-end. The *T_1ρ_* values of CH_3_ varied and were observed to be longer for dialkoxy groups with a longer chain length. CH_3_ from the additional alkoxy groups had additional mobility arising from internal rotation. Consequently, the mobility determined by the *T_1ρ_* values differed according to the length of the alkoxy substituents. The dialkoxy groups of HQ-TLCP have higher mobility than those of Naph-TLCP.

## Figures and Tables

**Figure 1 polymers-11-00992-f001:**
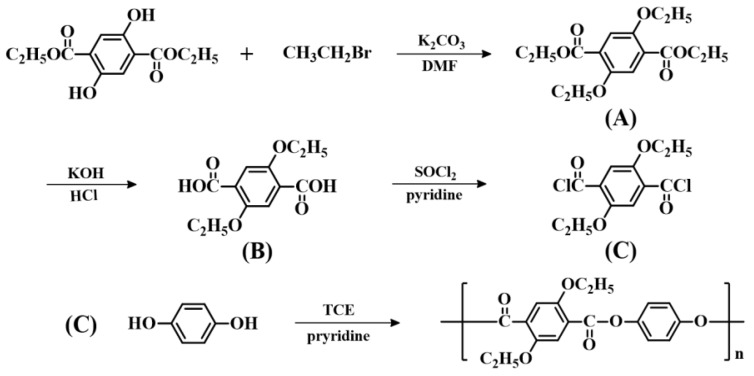
Synthetic routes of the thermotropic liquid-crystalline polymers (TLCPs).

**Figure 2 polymers-11-00992-f002:**
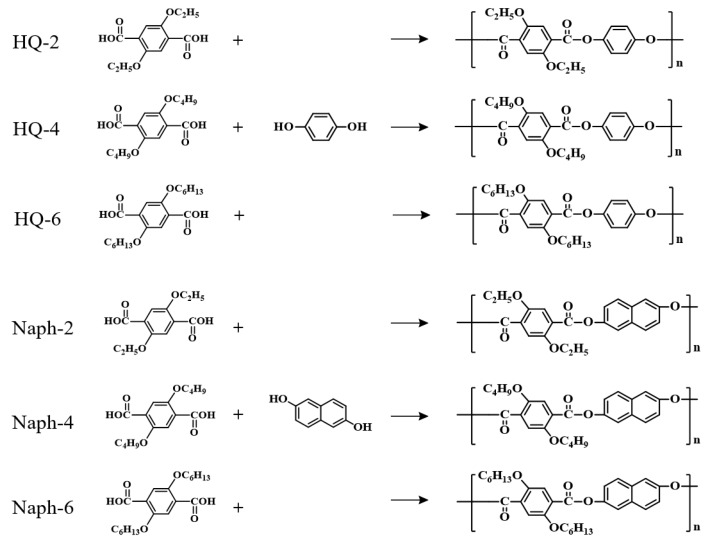
Chemical structures of hydroquinone (HQ)- and Naph-TLCPs.

**Figure 3 polymers-11-00992-f003:**
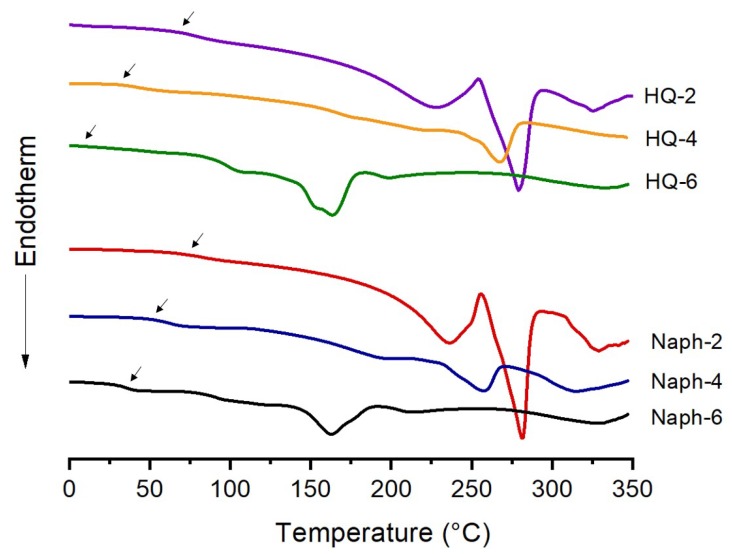
Differential scanning calorimetry (DSC) thermograms of HQ- and Naph-TLCPs.

**Figure 4 polymers-11-00992-f004:**
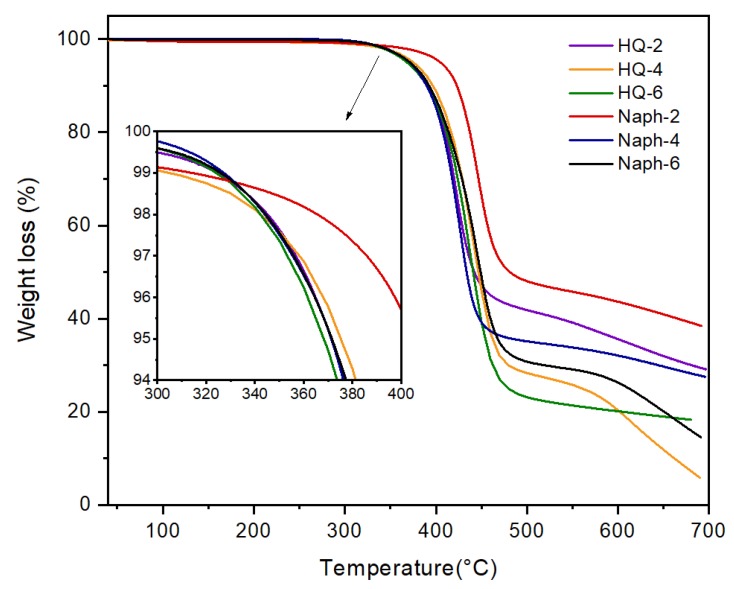
Thermogravimetric analysis (TGA) thermograms of HQ- and Naph-TLCPs.

**Figure 5 polymers-11-00992-f005:**
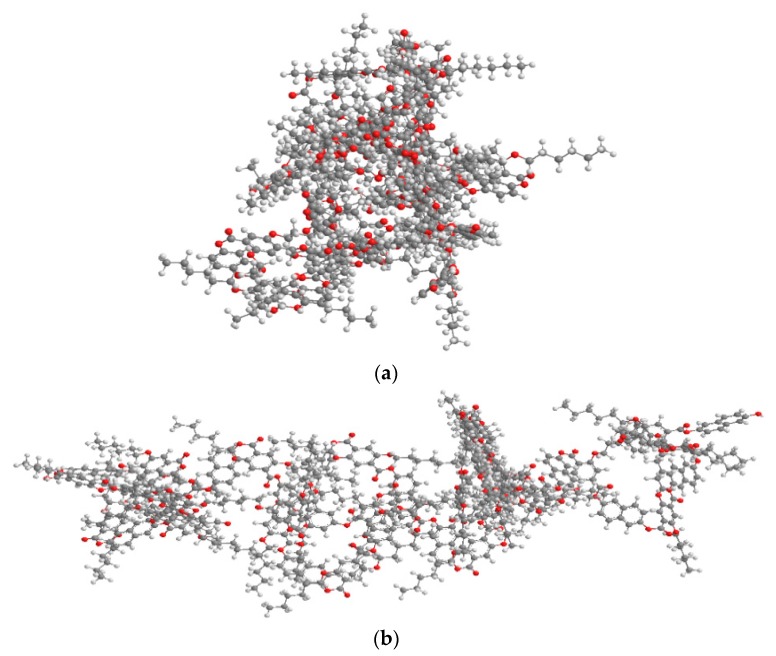
Comparison of the 3D chemical structures of (**a**) HQ-6 and (**b**) Naph-6.

**Figure 6 polymers-11-00992-f006:**
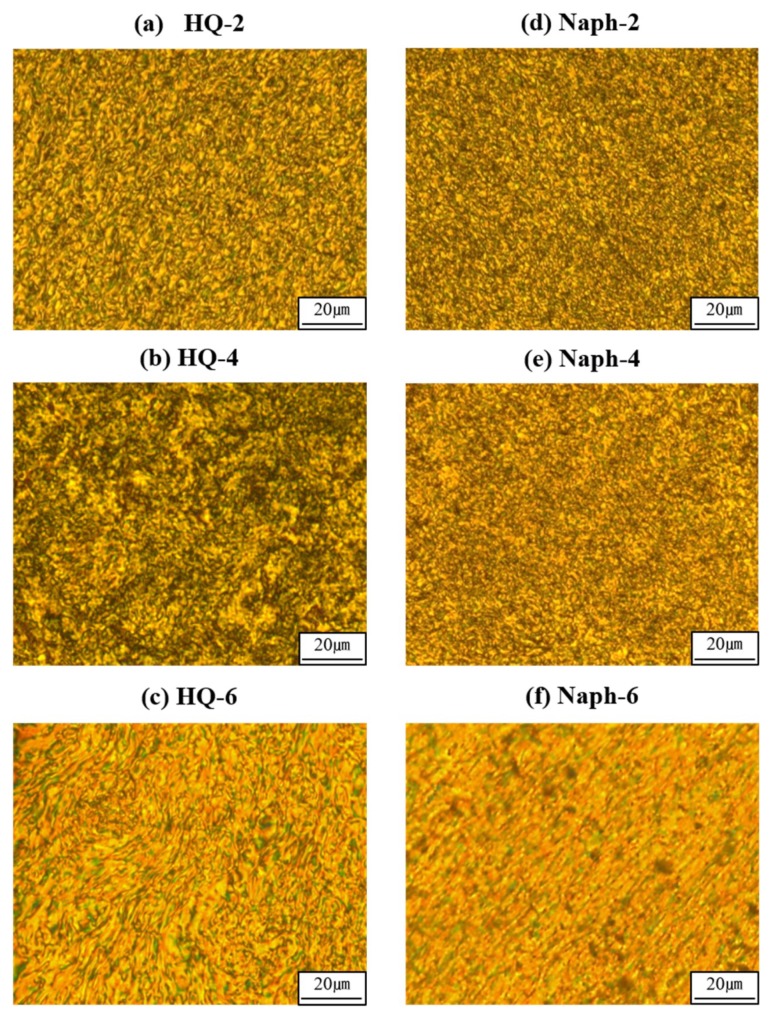
Polarized optical micrographs of (**a**) HQ-2 at 310 °C, (**b**) HQ-4 at 255 °C, (**c**) HQ-6 at 190 °C, (**d**) Naph-2 at 310 °C, (**e**) Naph-4 at 290 °C, and (**f**) Naph-6 at 200 °C (magnification 100×).

**Figure 7 polymers-11-00992-f007:**
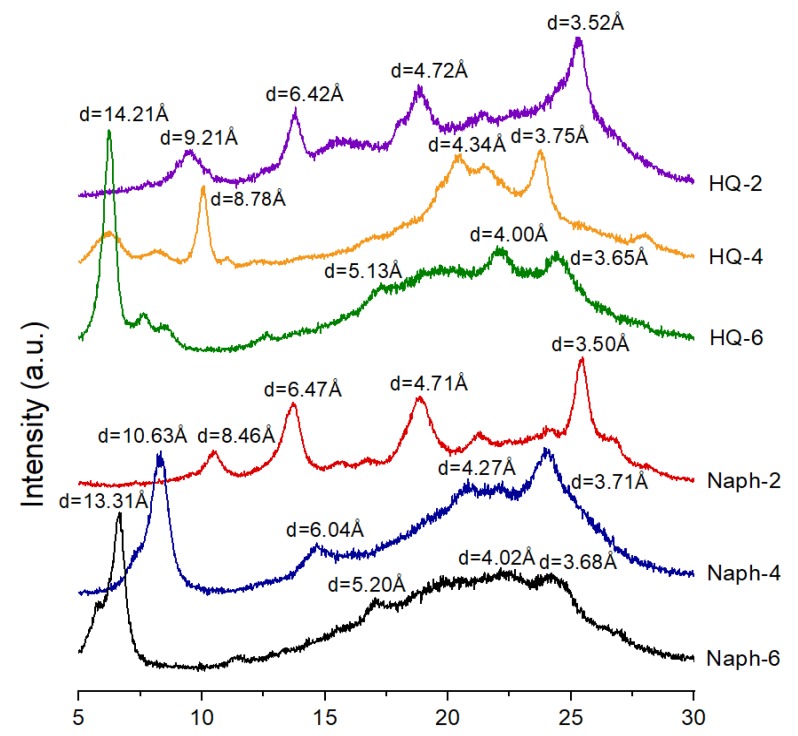
XRD patterns of HQ- and Naph-TLCPs.

**Figure 8 polymers-11-00992-f008:**
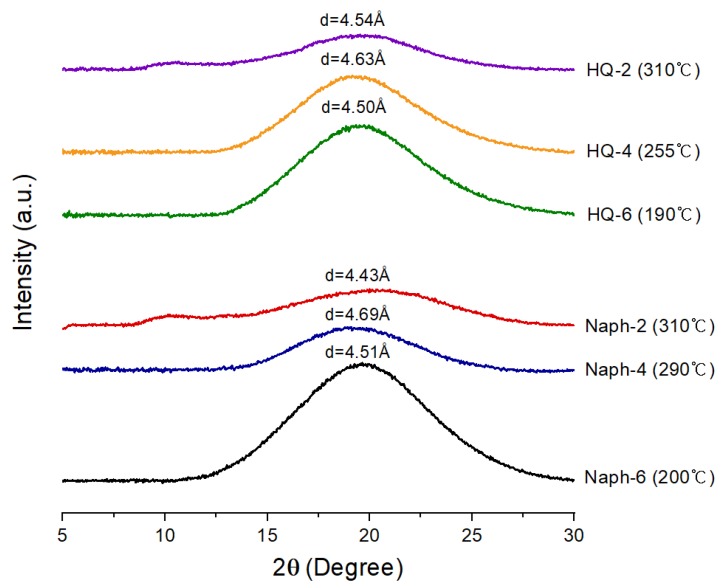
XRD patterns of HQ- and Naph-TLCPs in the LC mesogenic temperature ranges. The temperature in parentheses is the temperature at which the XRD patterns were collected.

**Figure 9 polymers-11-00992-f009:**
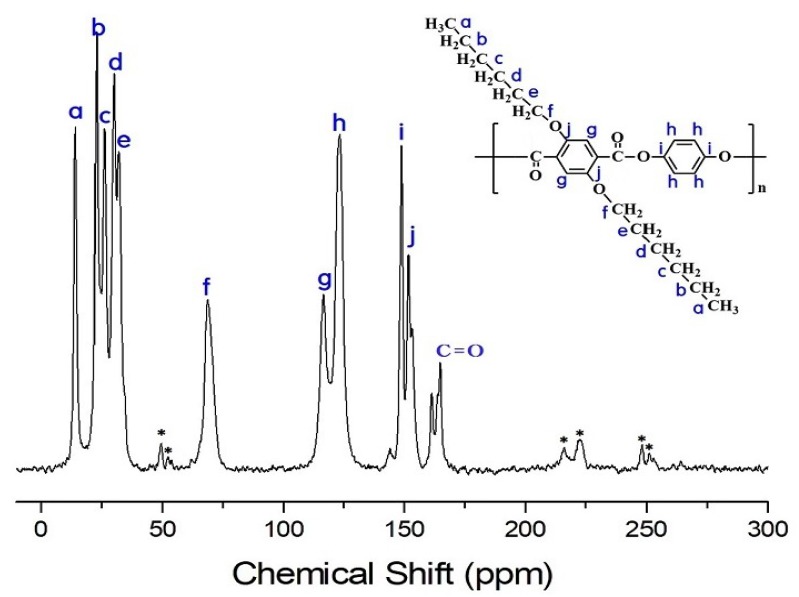
Solid state ^13^C CP/MAS NMR spectrum of HQ-6 at room temperature. The spinning sidebands are marked with asterisks.

**Figure 10 polymers-11-00992-f010:**
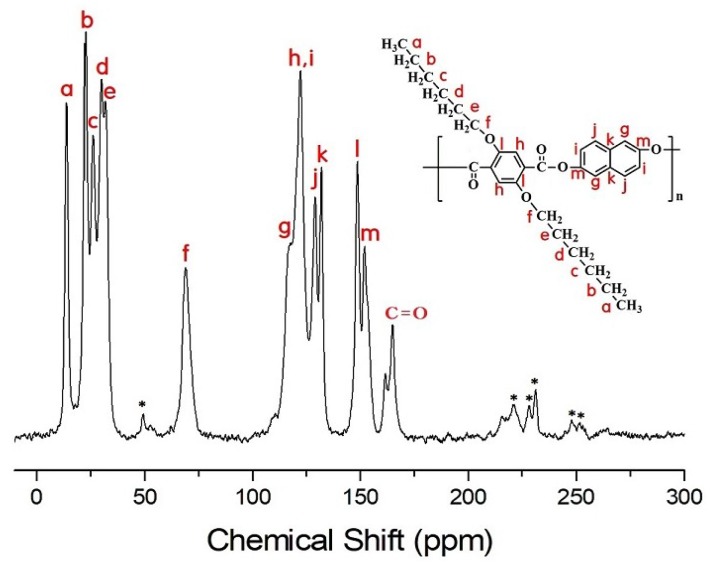
Solid state ^13^C CP/MAS NMR spectrum of Naph-6 at room temperature. The spinning sidebands are marked with asterisks.

**Table 1 polymers-11-00992-t001:** General properties of hydroquinone (HQ)- and Naph- thermotropic liquid-crystalline polymers (TLCPs).

TLCP	IV ^1^	*T_g_* (°C)	*T_m_* (°C)	*T_i_* (°C)	*T*_D_^i 2^ (°C)	wt_R_^600^ (%) ^3^	LC Phase	DC ^4^ (%)
HQ-2	0.99	71	279	325	357	27	Nematic	25
HQ-4	0.83	34	249	269	343	21	Nematic	19
HQ-6	Insol. ^5^	11	164	200	342	20	Nematic	18
Naph-2	0.95	76	282	329	365	48	Nematic	30
Naph-4	Insol.	54	258	312	362	34	Nematic	24
Naph-6	Insol.	38	164	227	348	26	Nematic	20

^1^ Inherent viscosity was measured at a concentration of 0.1 g/dL in phenol/p-chlorophenol/TCE = 25/40/35 (w/w/w) at 25 °C. ^2^ 2% initial weight-loss temperature. ^3^ Weight percent of the residue at 600 °C. ^4^ Degree of crystallinity. ^5^ Insoluble.

**Table 2 polymers-11-00992-t002:** *d* values corresponding to the XRD peaks of HQ- and Naph-TLCPs.

TLCP	Distance (Å)			
HQ-2	9.21 (9.6) ^1^	6.42 (13.8)	4.72 (18.8)	3.52 (25.28)
HQ-4	14.21 (6.22)	8.78 (10.08)	4.34 (20.48)	3.75 (23.74)
HQ-6	14.21 (6.22)	5.13 (17.28)	4.00 (22.2)	3.65 (24.38)
Naph-2	8.46 (10.46)	6.47 (13.68)	4.71 (18.86)	3.50 (25.46)
Naph-4	10.63 (8.32)	6.04 (14.66)	4.27 (20.82)	3.71 (23.98)
Naph-6	13.31 (6.64)	5.20 (17.04)	4.02 (22.14)	3.68 (24.20)

^1^ 2*θ* values are shown in parentheses.

**Table 3 polymers-11-00992-t003:** Spin-lattice relaxation time *T_1ρ_* (ms) in the rotating frame for each carbon in the TLCPs at room temperature.

TLCP	HQ-2	HQ-4	HQ-6	Naph-2	Naph-4	Naph-6
CH_3_-a	30.7	70.2	96.8	25.1	53.3	50.0
CH_2_-b	3.1	13.4	24.5	1.9	9.9	10.6
CH_2_-c		7.9	12.8		7.1	5.8
CH_2_-d		3.3	6.8		3.2	4.6
CH_2_-e			5.3			3.5
CH_2_-f			3.8			2.4
g	8.7	8.0	10.7	19.1	26.7	6.6
h	45.2	44.5	47.2	51.4	35.8	20.5
i	107.9	147.3	124.8			
j	105.1	115.2	108.5	120.2	106.1	149.9
k				51.2	38.8	96.6
l				158.4	140.5	96.1
m				102.6	125.8	69.1
C=O	118.5	163.3	144.7	179.3	193.4	104.7

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
