# Peer review of "Thermotropic Liquid Crystalline Polymers with Various Alkoxy Side Groups: Thermal Properties and Molecular Dynamics"

_polymers, 2019, doi:10.3390/polym11060992_

Round 1

Reviewer 1 Report

Dear autors,

The presented manuscript details the synthesis and characterization of two novel serires of thermotropic liquid crystalline polymers which are based on well known aromatic structures combined with aliphatic chains which helps to reduce the melting and processing temperatures of the polymers. The work is well structured and the methodology detailed is sound making the paper and the manuscript relevant to the readers.

However, the introduction briefly details the state of the art of previously studied TLCP which is not sufficient for readers who have no expertise in such materials. It would be interesting to extend the introduction, justify the innovation proposed in the manuscript and integrate potential applications of such materials.

The rest of the manuscript details the synthesis, characterization using different techniques. The interpretation of the results is aligned with the observations and most of the comments are adequate. However, some comparison with previous prepared materials is foreseen and will improve the quality of the manuscript.

My recommendation to the editor will be to accept the manuscript after the revision of the introduction and the discussion of the results.

With kind regards

Author Response

Dear Editor

This is my response to your comments regarding our paper “Thermotropic Liquid Crystalline Polymers with Various Alkoxy Side Groups: Thermal Properties and Molecular Dynamics” (polymers-509121) in polymers.

Thank you very much for the referee's comments. We have carefully revised the manuscript following the comments of the reviewer.

Reviewer-1 (Red-color in manuscript)

Q-1: However, the introduction briefly details the state of the art of previously studied TLCP which is not sufficient for readers who have no expertise in such materials. It would be interesting to extend the introduction, justify the innovation proposed in the manuscript and integrate potential applications of such materials.

A-1: See page 1, Introduction section. “Liquid crystals (LCs) are materials that exhibit partially ordered states, with their molecular order lying between those of three-dimensionally ordered crystals and disordered or isotropic liquids. As a consequence of the molecular order, LC phases are anisotropic, i.e., they have directional properties. Formation of LC states is a consequence of either structures with rigid-rod type units with a high axial ratio or of disc-like structures. These structural units are often called mesogenic units. The LC states of polymers were discovered with the discovery of aramids such as poly(p-phenylene terephthalamide) (Kevlar) and poly(p-benzamide) by DuPont de Nemours Co. in 1970s [1,2]. These aromatic polyamides form LC states when dissolved in a solvent (lyotropic) such as sulfuric acid. In addition, commercialization of aromatic polyesters (e.g. Xydar® and Vectra®) that form LC states in melts (thermotropic) in the 1980s has sparkled continued and unabated growth of the field of LC polymers (LCPs) [3,4].” was added.

Q-2: The rest of the manuscript details the synthesis, characterization using different techniques. The interpretation of the results is aligned with the observations and most of the comments are adequate. However, some comparison with previous prepared materials is foreseen and will improve the quality of the manuscript.

A-1: See page 2, Introduction section. “Wholly aromatic LCPs, especially when they are homopolymers, are highly crystalline, very-high melting, and very often interactive materials. Two representative examples are poly(p-hydroxybenzoate) and poly(p-phenylene terephthalate). These polymers have a very high melting temperature, and thus, they cannot be readily processed by melt spinning or injection molding. In contrast, copolyesters such as those (Xydar produced from Amoco, U.S.A.) prepared from p-HBA, TPA, and p,p'-biphenol are injection moldable at temperatures in the vicinity of 400 °C, which is not compatible with common melt spinning equipment. Thermal degradation of the copolyester at 400 °C, however, makes stable fiber production difficult. The other classes of wholly aromatic LCPs contain bent structural moieties such as resorcinol and bisphenol-A in addition to para-linked aromatic structures. Inclusion of asymmetrically substituted units such as chloro- or methylphenylene units can further depress the melting temperature. Further, the class of aliphatic-aromatic LCPs consists of mesogenic groups sequenced in a regular fashion with flexible, aliphatic segments of different lengths [18-20].” was added.

We hope this revision is satisfactory for your further process. 

Sincerely yours,

Jin-Hae Chang

Professor 

Kumoh Nat'l Inst. of Tech.

Reviewer 2 Report

This manuscript studies the thermal properties and molecular dynamic behaviors of new TLCPs from various dialkoxy terephthalate units with HQ or Naph. Typically, 13C cross‐polarization/magic angle spinning nNMR spectroscopy was utilized. The results are interesting, which can be accepted after considering the following possible revisions. 1. Why the carbon numbers of 2,3,6 were used in this paper? 2. Is there any odd-even effect in the present TLCP system? 3. In Fig 6, the data scale should be given. 4. In addition to the molecular structure which influence the LC behavior, the nanostructure (e.g., microphase separation) also plays an important role. The authors may consider citing some relevant references, Macromolecules, 2019, 52(4), 1892-1898; Angew. Chem. Int. Ed. 2018, 57(38), 12524-12528; ACS Appl. Mater. & Interfaces 2017, 9, 24846-24872 5. The XRD data of the TLCPs at their LC temperature should be important for assignment of their ordered structures. 6. The English writing needs improvement to meet with the publication standard.

Author Response

Dear Editor

This is my response to your comments regarding our paper “Thermotropic Liquid Crystalline Polymers with Various Alkoxy Side Groups: Thermal Properties and Molecular Dynamics” (polymers-509121) in polymers.

Thank you very much for the referee's comments. We have carefully revised the manuscript following the comments of the reviewer.

Reviewer-2 (Blue-color in manuscript)

Q-1: Why the carbon numbers of 2,4,6 were used in this paper?

A-1: When the length of the side group is changed, various changes in properties are observed. In this study, various physical properties were observed while varying the dialkoxy group length from 2 to 6.The characteristics of each TLCP were compared according to the variation in the side group length, with the number of carbons in the dialkoxy group varied as 2, 4, and 6.” was supplemented in Introduction section, in the middle of page 2.

Q-2: Is there any odd-even effect in the present TLCP system?

A-2: In this study, the change of alkoxy length to 2, 4, and 6 was observed, and the characteristics of the odd number were unobserved.

Q-3: In Fig. 6, the data scale should be given.

A-3: As the reviewer pointed out, the scale bars were supplemented in Figure 6.

Q-4: In addition to the molecular structure which influence the LC behavior, the nanostructure (e.g., microphase separation) also plays an important role. The authors may consider citing some relevant references, Macromolecules, 2019, 52(4), 1892-1898; Angew. Chem. Int. Ed. 2018, 57(38), 12524-12528; ACS Appl. Mater. & Interfaces 2017, 9, 24846-24872.

A-4: As the judges pointed out, we cited three references and supplemented the text with explanations. Introduction section, in the middle of page 2. In addition, various studies have been carried out using TLCPs having alkyl side chains. Recently, microphase separation was studied using an LC copolymer containing alkyl side groups [26-28].” was added.

Q-5: The XRD data of the TLCPs at their LC temperature should be important for assignment of their ordered structures.

A-5: As the reviewer pointed out, the XRD results obtained between Tm and Ti showing liquid crystal mesophases are shown in Fig. 8, and the explanation of Fig. 8 on page 10 is supplemented. See page 10, “In order to investigate the change in the morphological behavior, XRD patterns were obtained at each temperature showing the LC mesogenic phase (see Figure 8). The XRD patterns obtained between Tm and Ti are very different from those obtained at room temperature. That is, in the molten state, all the sharp peaks observed at room temperature disappeared and only a broad peak is observed. As a result, in the molten state, only an amorphous ordered structure formed.”

Q-6. The English writing needs improvement to meet with the publication standard.

A-6: As the reviewer pointed out, the entire manuscript of the English version was corrected by a specialized agency.

We hope this revision is satisfactory for your further process. 

Sincerely yours,

Jin-Hae Chang

Professor 

Kumoh Nat'l Inst. of Tech.
